# What Drives Different Governance Modes and Marketization Performance for Collective Commercial Construction Land in Rural China?

**Zhun Chen** [1,2], **Yuefei Zhuo** [3,]*, **Guan Li** [3] and **Zhongguo Xu** [3]

1   Department of Land Management, School of Public Affairs, Zhejiang University, Hangzhou 310058, China; 11422031@zju.edu.cn
2   Department of Environment and Planning, Henan University, Kaifeng 475004, China
3   Law School, Ningbo University, Ningbo 315211, China; liguan@nbu.edu.cn (G.L.); xuzhongguo@nbu.edu.cn (Z.X.)
*   Correspondence: zhuoyuefei@nbu.edu.cn

**Abstract:** The collective commercial construction land (CCCL) reform in China has attracted considerable attention worldwide, but studies on the influencing factors and performance of governance modes for CCCL marketization are still in their infancy. First, by deconstructing CCCL, this study developed a conceptual framework from the perspective of transaction cost economics. Based on a series of surveys, interviews, and closed questionnaires in two pilot areas, this study determined the influencing factors for governance mode choice for CCCL marketization through comparative case studies and compared the performance of the government-led and self-organized modes. This study concluded that asset specificity, uncertainty, and frequency were the main influencing factors for transaction costs, which could influence the choice of governance mode for CCCL marketization. Moreover, the characteristics of the two aforementioned governance modes, transaction costs, and specific revenue distribution resulted in different marketization performances.

**Keywords:** collective construction land; modes of governance; transaction attributes; transaction cost economics; performance difference

## 1. Introduction

With the rapid growth of China's economy and the acceleration of urbanization, land resources are becoming increasingly scarce [1]. China has produced a unique "dualistic structure of the land system" [2,3]. The land right in China is a public property right, which is different from the private property right in Western countries. In China, the property rights system of rural land is unique. On the one hand, land ownership of rural collective commercial construction land (CCCL) belongs to the collective [4]. The villagers' committee or village collective economic organization acts as the representative of the ownership subjects to exercise ownership [5]. On the other hand, to arouse the enthusiasm of users and improve the efficiency of land use, land-use rights are separated based on collective ownership. This research focused on the marketization reform of the CCCL, which transfers the use right of the CCCL under the premise of maintaining its collective ownership [6,7]. Under this background, the collective decision-making approach was formed. Specifically, during the CCCL transfer, the collective negotiates with users in the land market, thus exercising collective ownership. At the same time, public participation is necessary before the decision-making process [8]. The CCCL transfer can only be carried out with the consent of more than two-thirds of the villagers or their representatives. The local government allows for the rural collective economic organization to use CCCL, and users are permitted to trade the land-use right. Once the trade is completed, the land-use right of the CCCL is transferred from the collectives to the enterprises and individuals who are eager to use it. To address the problem of land resource shortage and promote

urban development, the Chinese government began selling construction land through bidding, auctions, and listings and expropriating land from collectives in rural areas at comparatively low prices [9,10]. In this way, land resource property rights changed from being collective-owned to state-owned. To a certain extent, the marketization reform of collective property rights of rural land increases the scale of rural land transfer and the income of the collectives and villagers, enhances the vitality the collective land market, and promotes economic development [11]. To use the state-owned construction land, the land users must pay many land-transferring fees (the revenue obtained by the local government when it trades the use right of construction land with land users). The value of the land-transferring fee depends on the land use type, location, and area of the plot, etc. Having the advantages of location, the price of construction land in urban areas is much higher than that in rural areas. The local government shares a large amount of money by charging land transfer fees. Land transfer fees increased from CNY 1.22 trillion to CNY 3.65 trillion during the period from 2007 to 2016 and it provided up to 36% of local government revenue [12]. However, compared with high land transfer fees, the compensation standard (the compensation for the farmers is 30 times the production value of agricultural crops) for villagers is extremely low [13]. As the scale of land resources is limited, it is used in urban areas as a priority. With the development of the economy, the scale of land resources provided for industrialization and urbanization is decreasing, which results in the increase in demand for land resources. As the scale of land resources is limited, the competition among land users becomes fierce, which leads to the increase in land value [14].

The land market is crucial for the development of market economies, the bonding of rural and urban land, and the promotion of rural–urban transformations and economic interactions [15,16]. A well-functioning land market will increase the productivity of land resources [17]. The CCCL marketization reform in China underwent three stages, namely forbidden circulation, invisible circulation, and standardized circulation [18]. Since the reform and opening-up in 1978, China's economy has developed rapidly, especially in the early 1990s. With the development of township enterprises and the rise of the rural housing boom, people's demand for collective construction land resources grew by the day. Moreover, resources were allocated through top-down, command-and-control administrative power. Under these circumstances, the land market and circulation of CCCL were lacking [19]. This absence eventually led to the spontaneous and disordered circulation of CCCL for a long time, that is, the aforementioned invisible circulation state [18]. According to the statistics [20], by the end of June 2000, 85% of the industrial land in Bao'an District, Shenzhen, was collective-owned construction land. Collective construction land in Shijie Town, Dongguan City, accounted for 66% of the 10 square kilometers of the built-up area. Half of the collective construction land was traded in Suzhou city, which covered an area of more than 100,000 mu (mu is a land size unit, where 1 mu = 0.066667 ha = 666.67 m$^2$). In addition, according to the municipal statistics of Huzhou City, from 1997 to 2002, a total of 334 cases of collective construction land were transferred, covering an area of 5562 mu, and the income was about CNY 58.58 million [21]. At the beginning of the 21st century, to regulate CCCL circulation, the Ministry of Land Administration of China began to implement CCCL marketization reform in cities such as Suzhou, Nanjing, Huzhou, Wuhu, and the Nanhai District [22]. The 17th Communist Party of China (CPC) Central Committee in 2008 and 18th CPC Central Committee in 2013 declared that an integrated urban and rural construction land market should be established, and rural CCCL should have the same development rights as state-owned land [23–25]. At the beginning of 2015, the National People's Congress authorized 15 pilot areas to conduct CCCL marketization reform [26]. At the end of August 2018, transactions for 10,100 CCCL plots covering an area of 90,531.50 mu were made (these data originated from our survey in the Ministry of Natural Resources and Planning). In a hierarchal administrative system, the local government has the final say about the allocation of resources. The marketization of CCCL tries to eliminate the local government's interference in the allocation of rural land

resources from the following three perspectives. First, the relevant policies and regulations of CCCL marketization point out that the local government is not allowed to interfere in the price mechanism. The price of CCCL is the result of the negotiation between the collective economic organization and land users and it is influenced by the vitality of the market, supply and demand of CCCL, and the local socioeconomic development level [27,28]. Second, all the collectives can apply for the trade of CCCL, and all the qualified land users can participate in the competition for the use right of CCCL [29]. There are diversified types of land that can be traded, such as industrial land, commercial land, warehouse land, education land, and scientific research land [17]. Like state-owned construction land, the CCCL can be traded by means of biddings, auctions, listing, trades, and the trade period varies according to the land use types and the negotiation between two parties. Compared with the first two trade means, the majority of CCCL are traded through listing [17]. Third, the land property rights are released through CCCL marketization [30]. There is no doubt that the use right of CCCL is released since it can be traded to land users. Even in some circumstances, the use right of CCCL can be transferred a second time. What is more, CCCL can be mortgaged. It is of great importance to rural revitalization as the collectives can use the money to make an investment, which results in the increase in both collectives' and villagers' income.

From the practice results of the CCCL marketization reform in the pilot areas, two types of governance modes emerged across the country. The first was the government-led mode and the other was the self-organized mode. The former mode reduces the possibility of opportunistic behaviors and negotiation costs from the holdout problem and the latter mode makes full use of connections within a village and reduces the possibility of conflicts and negotiation costs. These two important governance modes were adopted by most of the pilot areas, especially less-developed villages that were eager to enhance the income of their collectives. In the early stage, most of the pilot areas conducted marketization through the government-led mode [l]. With the development of the marketization reform, several pilot areas began to conduct marketization through the self-organized mode. Compared with the government-led mode, the self-organized mode can highlight the role of the market and the land marketization level is higher [31,32]. At the same time, from the perspective of CCCL marketization performance, the practice results differed (the performance of the marketization of CCCL refers to the physical outcome of the marketization, the distribution effect, and the process efficiency). For example, in some villages, through the marketization reform, infrastructure quality and the villagers' production and living standards improved. The collectives' and villagers' revenues increased to a considerable extent. In addition, the transaction period was short and the total cost was low. However, the marketization performance of other villages was poor. Specifically, their infrastructure quality was not improved, and the collectives' and villagers' revenues did not increase substantially. Furthermore, the transaction period was long and the costs were high [1].

In view of the above phenomenon, a series of problems emerged. Why did some pilot areas adopt the government-led mode, while others adopted the self-organized mode? What are the factors influencing the governance mode choice for CCCL marketization? Moreover, why does CCCL marketization performance differ in different regions? Is the difference caused by different natural resource endowments or socioeconomic development levels? In view of this, this research tries to answer two research questions. What are the factors influencing governance mode choice for CCCL marketization? What are the performance differences of the governance modes for CCCL marketization? Based on a literature review, studies on the influencing factors for, and performance differences in, governance modes for CCCL marketization seem to be in their infancy. The research on governance modes for CCCL marketization mainly focuses on two aspects. The first aspect is the effect of formal rules on CCCL governance modes [31,33,34]. The other aspect is the introduction and specific cases of different governance modes for CCCL marketization [32,33]. Research on governance modes for CCCL marketization can provide new insights into the CCCL reform. Thus, based on transaction cost economics (TCE)

and the alignment of transaction attributes and governance structures, this study aimed to conduct case studies to explain the influencing factors regarding selecting governance modes for CCCL marketization and compare the performance of the two governance modes. From the perspective of practice evidence, answering the two aforementioned research questions not only helped us to obtain a deeper understanding of the reason why diversified governance modes of CCCL marketization have emerged but it also provided a reference for CCCL marketization reform in other pilot areas regarding how to increase the marketization performance. From the perspective of the theoretical contribution, this study showed that the principle of alignment of transaction attributes with governance modes is true and TCE also applies in the public domain.

## 2. Conceptual Framework

To answer the two aforementioned research questions, a conceptual framework of the influencing factors for, and performance differences in, governance modes for CCCL marketization was constructed (Figure 1).

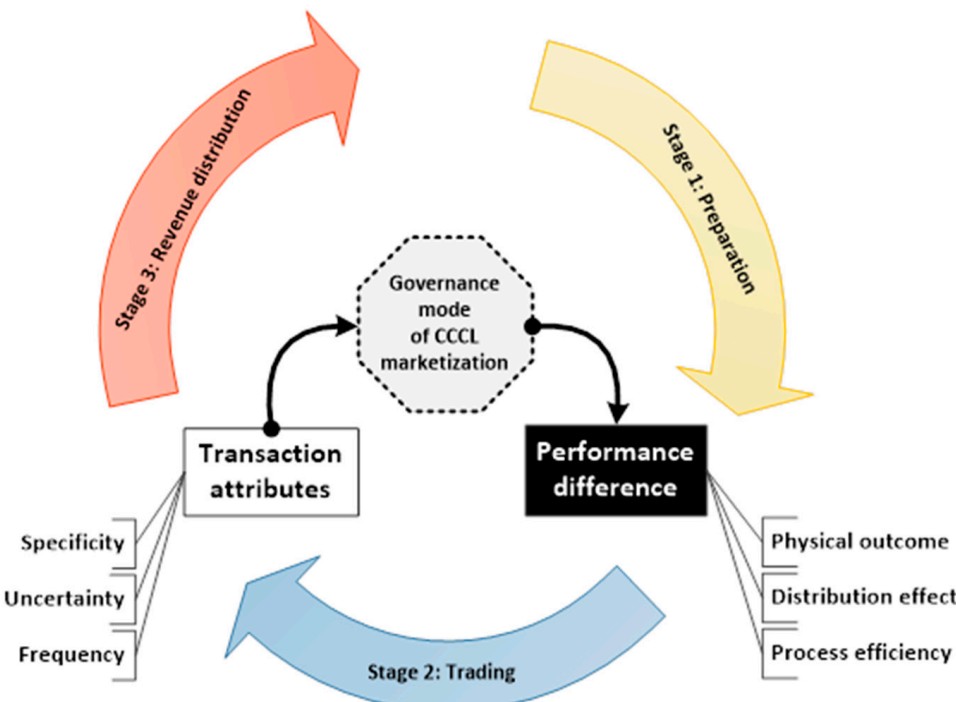

**Figure 1.** Conceptual framework. CCCL: collective commercial construction land.

In analyzing the alignment of governance modes with transaction attributes and its effects on transaction cost economization, land readjustment (Wang and Tan [1]) and rural renewal (Tan and Heerink [35]) were deconstructed into four stages. Similar to the two aforementioned studies, CCCL marketization can also be divided into three stages, namely the preparation stage, trading stage, and revenue distribution stage. In Figure 1, the arrows represent the three simplified CCCL marketization procedures. The dotted octagonal box in the middle is the governance mode of CCCL marketization. The rectangular box on the left represents the three classical transaction attributes. The black rectangular box on the right represents the three perspectives of performance differences.

In terms of the first research question, which originated from the alignment of transaction attributes with contract types, Williamson established the principle of the alignment of transaction attributes with governance structures [36]. According to Williamson, the governance structures take the forms of hierarchy, market, and hybrid [37,38]. The incentive functions and control of the market and hierarchy differ. In the market structure, when two parties make a deal, they react based on changes in prices. Thus, the market structure

provides strong incentives. However, as the parties in the transaction process in the market structure are engaged in a voluntary principle, the market structure is characterized by weak control. Therefore, the local government, which is a hierarchical governance structure, has an incentive to develop the economy [39,40]. The hybrid governance structure has the incentive effect of the market structure and the control effect of the hierarchical structure. In other words, the hybrid governance structure has semi-strong incentives and control. Meanwhile, the hierarchical structure has strong control and forces subjects to cooperate through coercion. In addition, the hierarchical governance structure has a certain degree of incentives. In the Chinese context, this structure demonstrates the characteristics of centralization in politics and decentralization in the economy. Only when the transaction attributes match the governance structures can transaction costs be reduced. For example, when the frequency is high, hierarchical and hybrid governance structures should be adopted because, compared with the large negotiation costs of the market, unified rules and top-down administrative power, which can reduce transaction costs [41,42].

Regarding the second research question, the performance differences in governance modes for CCCL marketization were analyzed from three perspectives, namely the physical outcome, distribution effect, and process efficiency. The physical outcome refers to the effects of CCCL marketization, such as an increase in collectives' and villagers' revenues, an improvement in infrastructures, and the provision of jobs for local villagers. The distribution effect refers to the revenue distribution among different stakeholders. For example, through CCCL marketization, villagers in some villages can obtain a sustainable increase in revenues, whereas those in other villages can obtain only a one-time revenue increase. Webster was the first to use the concept of process efficiency [43]. Process efficiency can be reflected in the CCCL transaction period, negotiation costs, administrative costs, conflict costs, information searching costs, land supervision costs, and transaction costs caused by information asymmetry and bounded rationality.

### 2.1. Transaction Stages of CCCL Marketization

TCE is a classical branch of new institutional economics that is studied by many distinguished scholars [44–46]. In Williamsonian transaction economics, a transaction is regarded as the basic analysis unit. According to Williamson, "A transaction occurs when a good or service is transferred across a technologically interface. One stage of activity terminates, and another begins" [36]. Based on this perspective, the entire CCCL marketization process can be regarded as a large transaction, that is, the transfer of rights and services, such as the use rights of CCCL that are transferred from a collective economic organization to a particular company or entrepreneur. Another example is the evaluation of land prices that are transferred from a specialized land appraisal company to collectives. Based on the above analysis, this study deconstructed CCCL marketization into three transaction stages, that is, the preparation, trading, and revenue distribution stages.

In the preparation stage, first, the collective economic organization surveys the CCCL area and location in the village. Second, local government staff members and the village committee negotiate with stakeholders for their unanimous agreement. The village collective economic organization holds meetings to discuss the details of the CCCL marketization procedure. Third, based on land-use planning and village development planning regulations, the scale of the CCCL, and the socioeconomic development level, the village collective economic organization chooses the location and governance mode for CCCL marketization. Finally, the village collective economic organization hires a professional company to complete the development, reclamation, and consolidation of the plot. In the trading stage, the village collective economic organization uses a rural public resources trading center to trade the CCCL through biddings, auctions, and listings. First, the village collective economic organization submits a marketization application to the Ministry of Natural Resources, the Committee of Development and Reform, the Bureau of Environmental Protection, the Bureau of Agricultural Management, the Committee of Economy and Information, etc., which are responsible for approving the application. Owing to the

information asymmetry between the superiors and subordinates and the bargaining between different administrative departments, the approval period is relatively long. Second, the collective economic organization entrusts the rural public resources trading center to publish detailed information on the land to be traded on its website. Subsequently, enterprises and individuals will compete for the use rights of the CCCL in biddings, listings, and auctions. Finally, the collective economic organization signs a contract with an enterprise or individual, and the rural public resources trading center publishes the result of the CCCL trade. In the revenue distribution stage, the local government, collectives, and villagers share the revenue from CCCL marketization. The local government receives a certain proportion of the land incremental revenue, which is determined by the socioeconomic development level of the area and use and location of the plot. Collectives typically use the revenue to make investments, such as purchasing government bonds, and villagers share the revenue based on their stocks. Occasionally, the revenue is evenly distributed among the villagers. The revenue distribution results differ between different governance modes.

*2.2. Transaction Attributes of CCCL Marketization*

Williamson identified three basic transaction attributes, namely specificity, uncertainty, and frequency [47]. Williamson established the principle of the alignment of governance modes with transaction attributes, which means that only when transaction attributes align with governance structures can transaction costs be reduced [37]. Specificity refers to the extent to which the assets used for certain activities can be reused in other fields without sacrificing the productive value [38]. Specificity can be divided into three categories, namely site specificity, physical asset specificity, and human capital specificity. Theoretically, as asset specificity increases, all investments, such as time, money, and staff, will face high risks of becoming sunk costs. Therefore, the lock-in effect increases, which can lead to opportunistic behaviors, such as holdout problems [1,41]. The larger the area, the more human capital and material capital invested and the higher the asset specificity [1,35]. For CCCL marketization, site specificity means that the land can only be used based on the general land use planning, urban and rural planning, and village development planning regulations. Physical asset specificity means that a substantial amount of money and equipment is necessary to obtain land-use rights and land consolidation. All costs face the risk of the lock-in effect and becoming sunk costs. Finally, human capital specificity means that professional experts are needed to ensure the completion of CCCL marketization.

Uncertainty can result from the limitations of information on certain transactions obtained by a party in the market and information asymmetry between superiors and subordinates [35]. The higher the degree of uncertainty, the higher the costs of the parties to guarantee the success of the transaction. Uncertainty can be further divided into two categories. Uncertainty I result from people's innate cognitive limitations and refer to the uncertainty caused by people's bounded rationality [48]. The greater this uncertainty, the more information that is needed to make decisions [41]. Uncertainty II is concerned with implementation and refers to the fact that the outcomes caused by certain actions are unknown [48]. When uncertainty II increases, the desire to adapt to disturbances also increases and may lead to maladaptation [47].

Frequency refers to the extent to which transactions are repeated [35]. From this perspective, transactions can be classified into three categories: one-time transactions, occasional transactions, and recurrent transactions. However, only a few transactions have completely isolated and discrete characters and can be considered as one-time transactions [49]. As the frequency of transactions increases, parties must repeatedly deal with property problems, which can increase the transaction costs. The frequency of CCCL marketization refers to the negotiations between the government staff and villagers to obtain the full support of the villagers before marketization and a fair share of the incremental revenue from the land after the transaction. When the transaction complexity increases, the coordination between different parties becomes difficult [50]. Based on the previous

decomposition of the CCCL marketization procedure, the main transaction attributes in the three transaction stages are shown in Table 1.

**Table 1.** Transaction attributes of the three transaction stages of CCCL marketization.

| Transaction Attributes | Specificity | Uncertainty I | Uncertainty II | Frequency | Complexity |
|---|---|---|---|---|---|
| Preparation | ✓ | ✓ | ✓ | ✓ | ✓ |
| Trading | ✓ | | ✓ | ✓ | |
| Revenue distribution | | | ✓ | ✓ | |

First, in the preparation stage, the main transaction attributes include asset specificity, uncertainty I, uncertainty II, complexity, and frequency. In this stage, asset specificity includes human capital specificity and material asset specificity. The development, reclamation, and consolidation of land require numerous personnel with professional knowledge and specialized equipment that are of little use in other fields. Second, the bounded rationality of the individuals designing the marketization scheme makes considering all the possibilities impossible, thereby leading to an imperfect marketization scheme. Therefore, this stage faces a substantial amount of uncertainty I. Moreover, when government staff and the village committee negotiate with stakeholders, information asymmetry and villagers' diverse preferences may result in a considerable amount of uncertainty II. In addition, either the village collective economic organization or specialized land consolidation company may engage in opportunistic behaviors, which can likewise lead to a substantial amount of uncertainty II. Third, the individuals designing the CCCL marketization scheme must consider numerous factors, such as the natural resource endowment, socioeconomic development level, land-use planning, and village development planning. Therefore, they face considerable complexity. Finally, to promote the CCCL, government staff and the village committee must negotiate with all stakeholders. They face high frequency—the larger the number of stakeholders, the higher the frequency.

Second, in the trading stage, the main transaction attributes include human capital specificity, uncertainty II, and frequency. Plot price evaluation, bidding, auction, and listing require personnel with professional knowledge who are of little use in other fields. Therefore, they face considerable human capital specificity. Moreover, this stage involves numerous procedures, and the natural resource endowment, socioeconomic development level, and CCCL demands differ from region to region. Thus, the transaction has a high possibility of failing. Therefore, it faces a substantial amount of uncertainty II. In addition, as the number of plots to be traded increases, the frequency also increases, which can increase the workload of the land evaluation employees and the tasks of the rural land resources trading center staff.

Third, in the revenue distribution stage, the main transaction attributes include frequency and uncertainty II. Each stakeholder wants to maximize his/her benefits. Therefore, some stakeholders may not be satisfied with the current revenue distribution and thus, demand revenue redistribution. Government staff and the village committee must negotiate with each stakeholder. Therefore, the larger the number of stakeholders, the higher the frequency. As the negotiation result depends on the game power of the above two parties, it faces a considerable amount of uncertainty II.

## 3. Materials and Methods

### 3.1. Study Area

Baiyun Village is located in the Hongguang Subdistrict, Pi County, Chengdu City, Sichuan Province. The area of Baiyun Village is 1.72 km$^2$, where the areas of the collective construction land and the agricultural land are 436.7 and 1955.85 mu, respectively. It has eight communities and the total number of residents is 1805, with an average gross personal income of 14,000 CNY/year. Huilong Village is located in Meijiang Subdistrict, Meitan County, Zunyi City, Guizhou Province. The area of Baiyun Village is 12.5 km$^2$,

where the area of agricultural land is 4500 mu. It has seven communities, with a total of 3468 residents.

The two typical governance mode cases for CCCL marketization were selected based on the typical case selection method proposed by Gerring [51]. Before we selected the two aforementioned cases, with the help of the staff in the Chinese Institute of Land Survey and Planning, we had the chance to study the cases of almost all 33 pilot areas. After studying all the cases, we found that the total number of cases was 438. The number of cases of the government-led mode was 306, which accounted for 70%. The number of cases of the self-organized mode was 132, which accounted for 30%. The government-led mode cases shared the following similarities. First, the local government promoted CCCL marketization actively. Second, the period of negotiation with stakeholders was relatively short. Third, the money needed for CCCL marketization was provided by the local government. Fourth, the local government shared a large proportion of the revenue. When selecting between the government-led mode cases, the characteristics of Huilong Village covered all the characteristics of government-led mode cases. Therefore, we selected Huilong as an example of a village using the government-led mode. The self-organized mode cases shared the following similarities. First, the leaders and elites in the village were responsible for promoting CCCL marketization. Second, the period of negotiation with stakeholders was relatively long. Third, the money needed for CCCL marketization was mainly provided by the elites of the village. Fourth, the collectives shared a large proportion of the revenue. When selecting between the government-led mode cases, the characteristics of Baiyun Village covered all the characteristics of self-organized mode cases. Therefore, we selected Baiyun as an example of a village that used the self-organized mode.

We conducted a series of surveys in Baiyun Village, Sichuan Province, in 2016, and in Huilong Village, Guizhou Province, in 2017. The former used a typical self-organized mode, and the latter employed a typical government-led mode. The government-led mode was the dominant mode employed in most of the pilot areas. Compared with the government-led mode, the self-organized mode was a newer governance structure, which was not adopted in many pilot areas. The two aforementioned cases were comparable to each other from the following two perspectives. First, the areas of the two traded CCCL plots were similar. The area of the CCCL plot in Huilong village was 19.78 mu and that of Baiyun village was 19.91 mu. Second, the location of Meitan County is close to that of Pi county, which eliminated the influence of natural resource endowment and socioeconomic development level on the choice of governance modes and the performances of CCCL marketization. The CCCL marketization performances of the two cases were excellent, and the two governance modes emerged and coexisted with each other over a long period, which made it interesting to figure out the influencing factors and performance differences for the two governance modes regarding CCCL marketization.

Differences in CCCL scale, socioeconomic development level, and historical problems mean that no one-size-fits-all governance mode exists for CCCL marketization, and each pilot area must choose a governance mode that matches the local conditions. The government-led and self-organized modes are two common governance modes for CCCL marketization. The government-led mode indicates that the entire CCCL marketization process is conducted and guided by the local government. The advantage of the government-led mode is that it can use administrative power to force stakeholders to cooperate, which can shorten the CCCL transaction period and improve the efficiency of the CCCL marketization process. In the early stage of CCCL marketization, the government-led mode was adopted by most collective economic organizations. According to Williamson's classification of governance structures, the government-led mode belongs under the hierarchical structure. As CCCL marketization progressed, by summarizing experiences in promoting CCCL marketization in the early stage, some villages with a high socioeconomic development level began to explore new governance modes for CCCL marketization. In contrast to the government-led mode, the self-organized mode indicates that the rural collective economic organization is responsible for specific CCCL marketization tasks, which are not interfered with by

the local government. For example, in the preparation stage, the village committee is responsible for negotiating with stakeholders. Moreover, the village collective economic organization is in charge of hiring a specialized company to complete the development, reclamation, and consolidation of the plots. In the trading stage, the collective economic organization, enterprises, and individuals search for trade information. In the revenue distribution stage, the collective economic organization has the final say about how the revenue will be distributed among the different stakeholders. The self-organized mode fully utilizes the leadership of the village elites to solve various problems during the CCCL marketization process and improve the efficiency of the marketization process. According to Williamson's classification of governance structures, the self-organized mode belongs under the hybrid structure. Figure 2 shows the locations of the two case study areas.

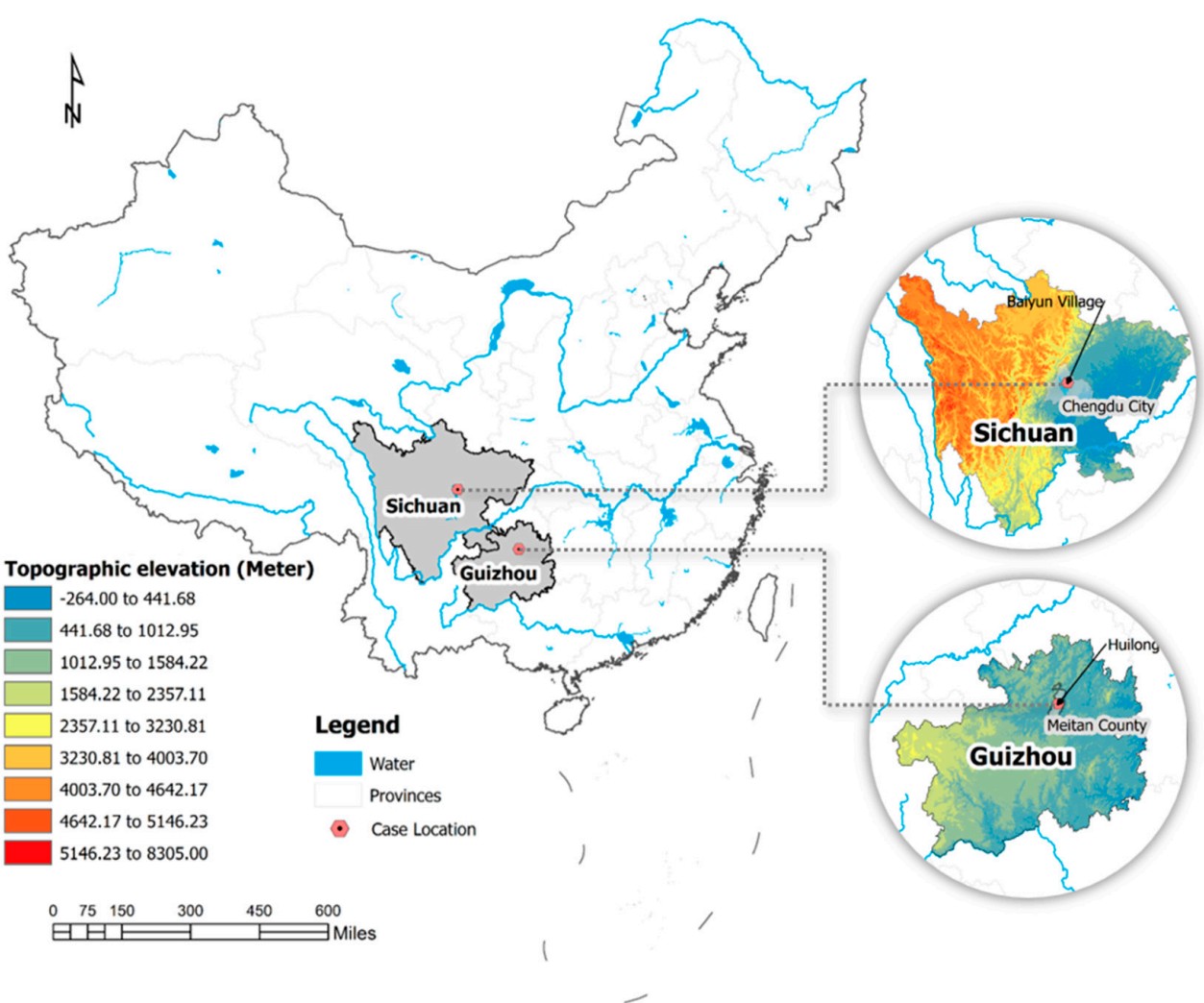

**Figure 2.** The locations of the two case study areas.

### 3.2. Survey Scheme

The surveys were funded and led by the Chinese Institute of Land Survey and Planning, which is a public unit under the Ministry of Land and Resources. The survey scheme consisted of three specific procedures. First, in the preparatory stage, we organized surveyors and collected relevant information on the study areas. Before we conducted the study, with the guidance of a tutor, our research group held several meetings to discuss the detailed survey scheme and analyze its feasibility. We consulted the officials in the Bureau of Natural Resources and Planning in Meitan County and Pi County and obtained

their permission to conduct the survey. Second, in the field survey stage, we visited the Bureau of Natural Resources and Planning to interview the government officials who were in charge of the CCCL marketization reform, villagers, and land users (enterprises and individuals) of the two case areas. The government officials introduced us to some marketization policies and regulations, recommended some successful CCCL marketization reform cases, and illustrated the commonly adopted governance modes and their characteristics for CCCL marketization in the two aforementioned case study areas. As village leaders are more familiar with the specific cases, they informed us of the details of the marketization cases such that we could collect information about the physical outcome and all categories of transaction costs produced during CCCL marketization. We visited land users to obtain information about the physical outcome and information searching costs for CCCL marketization. For the villagers, we concentrated on the performance of CCCL marketization from the perspectives of the physical outcome, process efficiency, and revenue distribution. Third, in the data compilation stage, we summarized the interview and questionnaire and formed case materials. Figure 3 illustrates the survey scheme.

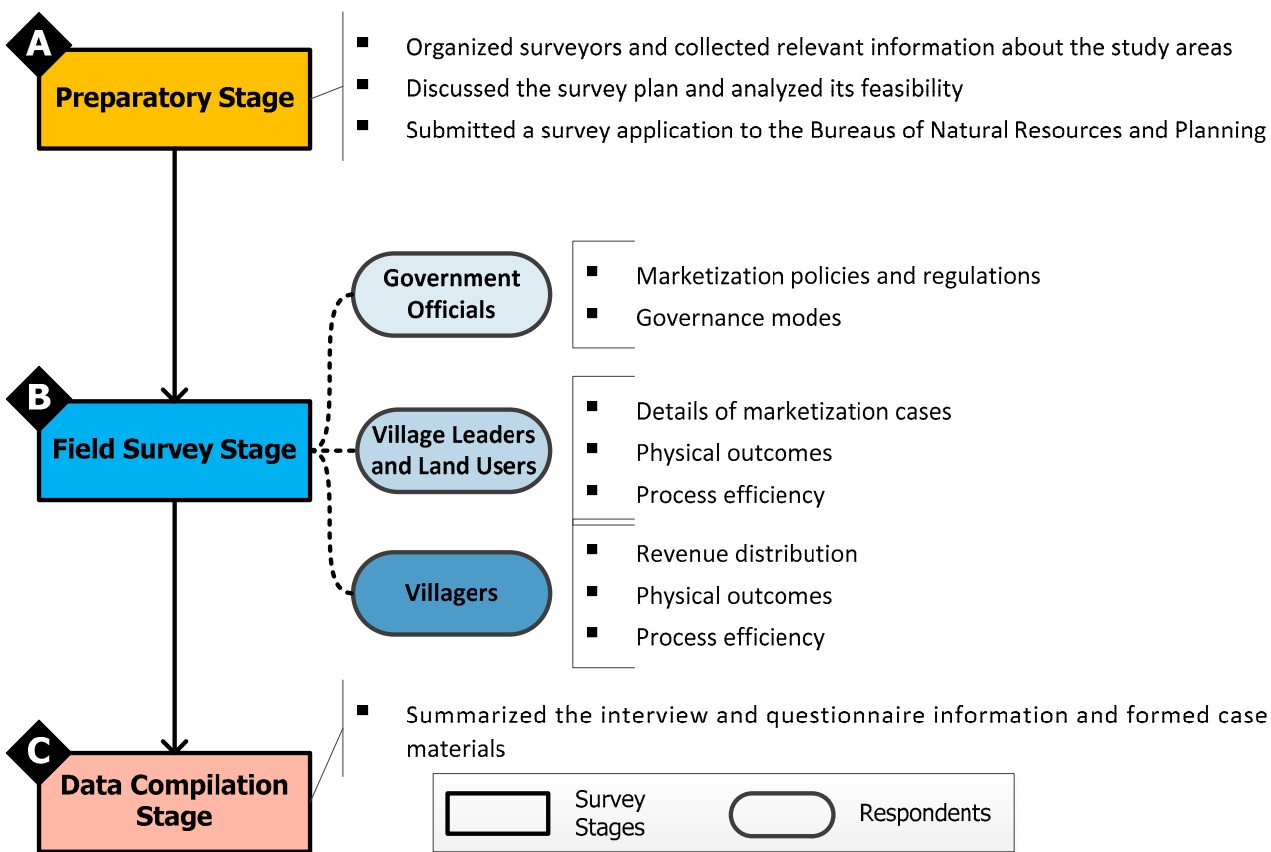

**Figure 3.** The survey scheme.

### 3.3. Interview and Questionnaire Scheme

We conducted face-to-face interviews and used a stakeholder-based method to select the interviewees [52]. The specific interview varies according to different respondents. We visited the Bureau of Natural Resources and Planning and the two selected villages to interview government officials, village leaders, land users, and villagers in order to obtain detailed information about CCCL marketization. The interviews varied between respondents. Before the interview, we prepared many questions that we were interested in asking and conducted the interview within our research group to consider whether the interview needed modification. Plans B and C were also designed, and we improvised

according to the answers of the respondents. The interview and questionnaire scheme are shown in Tables 2–4.

**Table 2.** The interview, questionnaire, and variables for government officials.

| Interview | Questionnaire | Variables |
|---|---|---|
| The relevant policies and regulations of the CCCL marketization | ◆ How many policies and regulations were carried out in this pilot area?<br>◆ What were the differences between the policies and regulations at the local government level and those at the central government level?<br>◆ How long was the period during which the relevant policies and regulations were designed? How often were the meetings held? | ◆ Process efficiency<br>  • Human capital costs, administrative costs, time costs |
| Introduction of governance modes for the CCCL marketization | ◆ How many kinds of governance modes for the CCCL marketization were there in this pilot area? Which one was the dominant governance mode? Why?<br>◆ What were the characteristics of the governance modes for the CCCL marketization?<br>◆ What were the costs of conducting the CCCL marketization using these governance modes? | ◆ Process efficiency<br>  • Negotiation costs, conflict costs |
| Process efficiency of the CCCL marketization | ◆ Were there any decision deficiencies? | ◆ Process efficiency |
| Revenue distribution among the different stakeholders | ◆ How was the marketization revenue distributed?<br>◆ What was the local government's share of the total revenue? | ◆ Performance<br>  • Revenue distribution |

**Table 3.** The interview, questionnaire, and variables for village leaders.

| Interview | Questionnaire | Variables |
|---|---|---|
| The details of the two cases | ◆ Preparation stage<br>  • What was the area of the CCCL plot? How many staff were involved in the marketization?<br>  • Have you ever felt regret after the marketization because you could have done it better?<br>  • Were there any difficulties when negotiating with stakeholders?<br>  • How many stakeholders were involved?<br>  • Were the stakeholders' preferences homogeneous? If no, how did you deal with heterogeneous stakeholder preferences?<br>◆ Trading stage<br>  • How many staff with professional and specialized knowledge were involved in the marketization?<br>  • How was the vitality of the CCCL market? What did it cost to outsource the land development and land consolidation project?<br>◆ Revenue distribution<br>  • Were there any stakeholders who disagreed with the revenue distribution? If yes, how did you deal with this situation? | ◆ Preparation stage<br>  • Specificity<br>  • Uncertainty I<br>  • Uncertainty II<br>  • Frequency<br>  • Complexity<br>◆ Trading stage<br>  • Human capital specificity<br>  • Uncertainty II<br>◆ Revenue distribution<br>  • Uncertainty II<br>  • Frequency |
| Physical outcome of the CCCL marketization | ◆ Did the collective income increase after the marketization?<br>◆ Was the infrastructure improved after the marketization? | ◆ Performance<br>  • Physical outcome |

**Table 3.** *Cont.*

| Interview | Questionnaire | Variables |
|---|---|---|
| Process efficiency of the CCCL marketization | ◆ How long was the period of negotiating with the stakeholders? <br> ◆ How long was the period of the examination and approval of the submitted materials? <br> ◆ How long was the period of regulating the use of the CCCL? <br> ◆ How long did it take to find enterprises and individuals who were willing to use the CCCL? | ◆ Process efficiency <br> • Negotiation costs <br> • Administrative costs <br> • Regulation costs <br> • Information searching costs |
| Revenue distribution among the different stakeholders | ◆ What was the collectives' share of the total revenue? <br> ◆ How did the collectives use the revenue? | ◆ Performance <br> • Revenue distribution |

**Table 4.** The interview, questionnaire, and variables for the villagers and land users.

| Respondent | Interview | Questionnaire | Variables |
|---|---|---|---|
| Villagers | Physical outcome of the CCCL marketization | ◆ Did the average personal income increase after the marketization? <br> ◆ How many stakeholders were resettled? <br> ◆ What was the compensation standard for the stakeholders? <br> ◆ Through the CCCL marketization, how many villagers were employed? | ◆ Performance <br> • Physical outcome |
| | Process efficiency of the CCCL marketization | ◆ Were there any villagers who disagreed with the marketization and thought that it would finally lead to a holdout problem? | ◆ Process efficiency <br> • Negotiation costs |
| | Revenue distribution among the different stakeholders | ◆ What was the villagers' share of the total revenue? <br> ◆ How did villagers use the revenue? | ◆ Performance <br> • Revenue distribution |
| Land users | Physical outcome of the CCCL marketization | ◆ What did enterprises do to help local villagers? | ◆ Performance <br> • Physical outcome |
| | Process efficiency of the CCCL marketization | ◆ How long did it take to find collective economic organizations who were willing to transfer the use right of the CCCL? | ◆ Process efficiency <br> • Information searching costs |

For government officials, we concentrated on the relevant policies and regulations of CCCL marketization, the introduction of the governance modes of CCCL marketization, the efficiency of the CCCL marketization process, and the revenue distribution of the local government. For village leaders, we mainly focused on the physical outcomes of CCCL marketization, the efficiency of the CCCL marketization process, and the revenue of the collectives. The interview with the village leaders was conducted from the perspective of transaction stages and the performance of CCCL marketization. For example, concerning the physical outcomes of CCCL marketization, we asked them about the compensation standard of resettlement, the quality of infrastructure, the total revenue of the CCCL marketization, and the revenue of the collectives. As for the process efficiency, we asked them about the period of negotiation with stakeholders, the times the village leaders held meetings, and the period during which relevant trade information was sought. Regarding the revenue distribution, we asked them about how revenue was distributed among different stakeholders, how the collectives used the revenue, etc. During our interview, we found that the collectives often shared most of the revenue to enhance the infrastructure and make an investment. Villagers only shared a small amount of revenue. Regarding land

users, we paid special attention to the opportunities they provided for local villagers and the information searching costs. Regarding villagers, we mainly focused on their conflicts with the staff in the village committee, hold-out problems, their share of revenue, and how they used that revenue.

In addition, to avoid misunderstandings and to verify the information, we used a closed questionnaire as a supplement to the interview. After we interviewed the aforementioned four categories of respondents, we asked them to fill in the questionnaire. The purpose of the questionnaire was to determine which variables we needed to analyze the factors influencing, and the performance of, the governance modes for CCCL marketization. For example, to compare the asset specificity of the two cases, we asked them for information about the area of the CCCL, the number of staff working in the administrative department and the rural public resource trade center, and the cost of outsourcing the land development and land consolidation project to a certain specialized company. Regarding uncertainty, we asked them for information about the difficulties that officials have when negotiating with the stakeholders, the GDP of Meitan County and Pi County, the average personal income, the vitality of the CCCL market, and the villagers' opinions about revenue distribution. Regarding frequency, we asked them for information about the scale of the stakeholders and the number of plots that have been traded. Regarding complexity, we asked them for information about the influencing factors that the individuals face when designing the CCCL marketization scheme, as well as the stakeholders' preferences. Like the interview, the questionnaire was also designed according to different respondents. For example, we asked government officials about the categories of policies and regulations, the differences between policies and regulations in the local area and those issued by the central government, the period during which the relevant policies and regulations were designed, how often meetings were held, and the characteristics and transaction costs of the governance modes for CCCL marketization. We asked village leaders for basic information about the specific CCCL marketization case, such as the ownership of the CCCL, the area and location of the plot, the enterprises and individuals who acquired the use right of the CCCL, the land-use type of the CCCL, the rental period, and the total revenue of the marketization. In addition, regarding both cases, we concentrated on the period of negotiation, the approval of submitted materials, the regulation of the use of the CCCL, and how land users were found. We asked land users about the beneficial policies they provided for the local villagers and the time it took to find the collective economic organization that was willing to transfer the use right of the CCCL. We asked villagers about the average personal income, the number of stakeholders, the compensation standard, the number of employed villagers, and the share of their revenue.

To avoid the differences in understanding of the same item, after we conducted the interview and questionnaire, we randomly selected different categories of respondents to cross-check the information both in the interview and questionnaire. For example, when asking the respondents to scale on the item, we invited all of them to gather in a meeting room. Then, we introduced them to the aim of the interview and questionnaire and explained the meaning of each item to make sure that they understood it. Through the survey, interview, and questionnaire, the following information was obtained. First, in the preparation stage, since the staff must reconcile with different preferences of the stakeholders, it took too much time to negotiate with them, which produced high negotiation costs. What is more, as the specificity of land development and the land consolidation project was high, the collective economic organization often outsourced it to a certain professional company. Second, in the trading stage, concerning the factors such as market vitality of CCCL, socioeconomic development level, and the supply and demand of CCCL that vary from area to area, the result of the trade of CCCL faced high uncertainty. Third, in the revenue distribution stage, the collectives often shared most of the revenue and they used them to enhance the infrastructure and make an investment. Compared with the revenue of the collectives, villagers only shared a little revenue.

## 4. Results

*4.1. Description and Characteristics of the CCCL Marketization Projects*

Baiyun Village, Chengdu City, Sichuan Province, is a typical self-organized mode case, and the processes of CCCL marketization were as follows. In the preparation stage, the number of land resources within the village was investigated and several meetings were held to determine the exact locations of the plots for consolidation. On 28 March 2013, Chengdu Baiyun founded the Collective Asset Management Co., Ltd. Next, the marketing scheme was designed based on the advice of experts, the desires of the villagers, the requirements of the industrial development plan, and the approval of the government at the township and county levels. Subsequently, the marketing scheme was approved by the Hongguang Town government and the Bureau of Land Resources of Pi County. In May 2013, the Duoli Company cooperated with the collective economic organization. The amount needed for the project was CNY 8 million, which was provided by the Duoli Company. The land consolidation project began in November 2015, where the number of stakeholders was 810. Finally, 330 mu of land was consolidated. In the trading stage, Chengdu Baiyun developed the Asset Management Co., Ltd. to apply to the public resources trading service center of Pi County and submit marketing documents. The trading center was responsible for the announcement and application. On 6 February 2016, the public resources trading service center of Pi County offered a listing for the land-use rights of four parcels of land. The Duoli Company won the listing on 11 March 2016 and signed a contract with the collective economic organization. The area of the four plots was 19.91 mu, which was to be used for commerce with a 40-year rental period. Based on the opinions of the villagers, the benchmark land price of the area, and the cost of the land consolidation, the initial listing price was estimated at 680,000 CNY/mu. In the revenue distribution stage, as the land was for commercial use and transacted via a listing, the plot was determined to be at level three of the benchmark land price. Therefore, the collective economic organization of Baiyun Village submitted 15% of the total revenue (CNY 13.54 million), which was CNY 2.03 million. The cost of the land reclamation and housing construction for the villagers was CNY 10.47 million, which meant that the collective economic organization and villagers obtained a share of CNY 1.04 million. The villagers received 20% and the collective economic organization received 80%. The villagers used their land contracting rights to form a land stock cooperative organization in which they could receive a certain amount of rent in addition to their share. The rent was 2275 CNY/mu. The land stock cooperative organization was contracted with Duoli Company in areas such as selling flowers and trees and providing laborers. Furthermore, the villagers obtained periodic shares.

Huilong village, Meitan County, Guizhou Province, is a typical government-led mode case and the processes of CCCL marketization were as follows. The socioeconomic development level of Huilong Village was high, and enterprises and individuals were eager to use CCCL. However, based on the general land use and development plans of the village, the scale of the CCCL in Huilong Village was limited and could not meet the needs of the villagers and enterprises. Although the socioeconomic development level of Luohuatun Village and Qunxing Village was relatively low, the scale of the CCCL in these two villages was relatively large. Therefore, the above two villages were motivated to increase the income of their village and villagers by trading their quota with Huilong Village. Guided by the Bureau of Natural Resources and Planning of Meitan County and the local government, the two villages began preparing for a trade with Huilong Village. In the preparation stage, the collectives of Luohuatun Village and Qunxing Village invited a professional company to design their land reclamation plan. The collectives of Huilong Village also invited a professional company to design their land development plan and choose the exact location of the plot. The village committee of the three villages negotiated with the stakeholders to obtain a unanimous agreement and paid compensations. The number of stakeholders was 2000. The collective economic organization of Huilong Village designed a marketization scheme and negotiated with Luohuatun Village and Qunxing

Village regarding the price of the quota. The three villages agreed to trade paddy fields at the price of 100,000 CNY/mu, dry land at the price of 80,000 CNY/mu, and other agricultural land at the price of 40,000 CNY/mu. Huilong Village purchased 16.33 mu of construction land from Luohuatun Village, including 7.68 mu of paddy fields and 8.65 mu of dry land, paying compensation of CNY 1.46 million. In addition, Huilong Village paid Qunxing Village CNY 234,800 for 2.42 mu of dry land and 1.03 mu of other agricultural land. In the trading stage, Huilong Village hired a professional company to evaluate the land price. Subsequently, the collectives requested the public resources trading center of Meitan County to provide the specific transaction information, such as the use, location, and price of the land, as well as the standards of qualified enterprises, on its website. In October 2015, two enterprises obtained the use rights of the CCCL, one of which was Tongxinyuan Commercial Co., Ltd. The area of the plot was 16.8 mu, and the land was to be used for storage. The average price was 171,800 CNY/mu and the listing period was 50 years. In addition, the total price was CNY 2.88 million. The other company was Lin Sheng Tea Co., Ltd. The area of the two plots of industrial land was 2.98 mu and the listing period was 50 years. Furthermore, the total price was CNY 510,000. In the revenue distribution stage, the total price of the marketization was CNY 3.39 million. The total cost of the marketization was CNY 2,624,800, which included the compensation for crops and land (CNY 720,000), the cost of the purchasing quota (CNY 1,694,800), the measurement and evaluation costs (CNY 20,000), and the insurance for the villagers who lost their land (CNY 190,000). The government charged 20% of the gross revenue, which was approximately CNY 153,000. The village collectives received 50% of the remaining revenue, which they used to enhance the basic infrastructure, and income, which was approximately CNY 306,100. Each villager in Huilong Village received CNY 92.3. By trading 16.33 mu of land, Luohuatun Village received CNY 1.46 million. The village collectives received 60% of the revenue, which was CNY 876,000. In addition, the villagers received 40% of the revenue, which was CNY 584,000. By trading 3.45 mu of land, Qunxing Village received CNY 234,800, and the village collectives received 60% of the revenue, which was CNY 149,900. The villagers received 40% of the revenue, which was CNY 93,900 in total.

The diversified CCCL governance modes provided a foundation for conducting the case studies. The comparison between the two cases was undertaken to help with optimizing governance modes for CCCL marketization and developing efficient and fair policies for marketing. This section focused on identifying and analyzing the alignment of the transaction attributes with the two governance modes and their marketization performances in the three transaction stages. The comparison between Baiyun Village and Huilong Village in terms of the degree of the transaction attributes in the three transaction stages is presented in Table 5.

**Table 5.** Comparison of the CCCL projects between Baiyun Village and Huilong Village in terms of the transaction attributes at different transaction stages.

| Transaction Stages/Transaction Attributes | Baiyun Village | Huilong Village |
|---|:---:|:---:|
| Preparation | | |
| ■ Asset specificity | + | ++ |
| ■ Uncertainty I | + | + |
| ■ Uncertainty II | + | ++ |
| ■ Complexity | ++ | + |
| ■ Frequency | + | ++ |
| Trading | | |
| ■ Human capital specificity | + | + |
| ■ Uncertainty II | ++ | + |
| Revenue distribution | | |
| ■ Uncertainty II | + | ++ |
| ■ Frequency | + | ++ |

Note: + denotes strong, ++ denotes very strong.



*4.2. Influencing Factors of Governance Modes for CCCL Marketization*

The preparation stage is the foundation for CCCL marketization. This stage mainly involves determining the location of the land for marketing, negotiating with villagers and obtaining a consensus, and designing the marketing strategy. This stage is particularly important to the local government, collective economic organization, and villagers, as it affects not only the industrial development of an area but also the revenue distribution. Obtaining villagers' full support is a precondition for CCCL marketing. The negotiation process may take years owing to property rights problems and other factors. The transaction in the preparation stage refers to the negotiation between government staff and villagers. The transaction attributes in this stage include uncertainty I, uncertainty II, complexity, and frequency. Specifically, first, the staff responsible for designing the marketing scheme cannot consider all the possibilities. Thus, the scheme may be incomplete. Second, when negotiating with villagers, owing to differences in their values and preferences, government staff will encounter communication difficulties, which can increase the uncertainty of the result of the negotiation. Third, owing to differences in natural resource endowments and the social and economic conditions of each pilot area, various factors should be considered before designing the scheme for formulating the market entry plan, which can increase the difficulty of designing the marketing scheme and complexity in this stage. Finally, frequency in this stage refers to the number of negotiations between government staff and villagers.

From the perspective of asset specificity, the larger the area of the plot, the more the human capital and material capital that was invested and the higher the asset specificity. The area of the plot in Huilong Village was 19.78 mu and that in Baiyun Village was 19.61 mu. Therefore, the asset specificity of Huilong Village was greater than that of Baiyun Village. From the perspective of uncertainty I, owing to the limitations of the cognitive ability of the individuals designing the marketization scheme, the suitability and feasibility of the marketization plan demonstrated considerable uncertainty. Therefore, the two regions experienced a substantial amount of uncertainty I. From the perspective of uncertainty II, the village committee of Baiyun Village had to negotiate with the stakeholders involved in the land consolidation area, and the committees of all three villages had to negotiate with all stakeholders. From the perspective of the scale of the stakeholders involved, Huilong Village included 2000 people and Baiyun Village included 810. The larger the scale of the stakeholders, the greater the difficulty of the village committee during the negotiation and the greater the uncertainty of the negotiation result. Moreover, the funds used for the land consolidation in Baiyun Village were derived from a company cooperating with the village. The two parties shared interests and risks, which made engaging in opportunistic behaviors impossible for either party. The funds used for the marketization of Huilong Village were mainly from the special funds appropriated by the local government. Given the complexity of the approval process of the funds and the long approval period, they faced considerable uncertainty. Therefore, overall, compared with Baiyun Village, Huilong Village faced a greater amount of uncertainty II. From the perspective of frequency, the larger the area of the plot, the larger the scale of the stakeholders and the higher the frequency. The area of the plot in Huilong Village was larger than that in Baiyun Village and the scale of the stakeholders involved in Huilong Village was larger than that involved in Baiyun Village. Therefore, the frequency in Huilong Village was higher than that in Baiyun Village. From the perspective of complexity, the collective economic organization of Baiyun Village had to complete the demolition of old houses and the construction of new houses. Meanwhile, the collective economic organization of Huilong Village only needed to finish the consolidation of land resources. Therefore, the complexity in Baiyun Village was higher than that in Huilong Village. Theoretically, when the complexity is high, adopting a decentralized governance structure is suitable. When asset specificity, uncertainty II, and frequency are high, a centralized governance structure should be adopted to reduce the transaction costs. In practice, Baiyun Village adopted the self-organized mode, whereas Huilong Village adopted the government-led mode, which is consistent with the theoretical analysis.

Human capital specificity and uncertainty II are the main transaction attributes in the trading stage. Staff with specialized knowledge are responsible for checking the materials relevant to the marketization and the bidding, auction, and listing of the CCCL use rights. The staff must be trained over a long period to deal with the work. The staff's knowledge and skills are not useful in other fields. Therefore, both villages faced high human capital specificity. Compared with human capital specificity, uncertainty II is the main transaction attribute in this stage. In this stage, uncertainty II refers to the uncertainty of the result of the marketization. It is highly possible that no enterprises will be eager to use the land, which can result in the failure of the plot transaction. The higher the socioeconomic development level, the larger the enterprises' need for the CCCL, the lower the possibility of the failure of the plot transaction, and the lower the uncertainty II. In 2016, the GDP of the Pidu District was CNY 46.27 billion and that of Meitan County was CNY 9.011 billion. Therefore, the uncertainty II in Huilong Village was higher than that in Baiyun Village. Theoretically, when uncertainty II is high, adopting a hierarchical structure or hybrid structure, which shares the characteristics of the hierarchical structure, is suitable. In practice, Huilong Village adopted the government-led mode to promote CCCL marketization, which is in accordance with the theoretical analysis.

Uncertainty II and frequency are the main transaction attributes in the revenue distribution stage. In this stage, uncertainty II refers to the uncertainty of the result of the revenue distribution. Although revenue distribution is determined in the preparation stage, some villagers may have different opinions regarding the matter. Villagers who are not satisfied with the result of the revenue distribution are eager to redistribute the revenue. Conflicts may arise between village committees and villagers during negotiations, which may lead to serious problems. Compared with Baiyun Village, the scale of the stakeholders involved in Huilong Village was larger. Therefore, the uncertainty II and frequency in Huilong Village were higher than those in Baiyun Village. When uncertainty II and the frequency are high, adopting a highly centralized governance structure is suitable. In practice, Huilong Village adopted the government-led mode, which verifies the theoretical analysis above.

### 4.3. Performance Differences

A comparison between Baiyun Village and Huilong Village in terms of performance differences is shown in Table 6.

For the case of Baiyun Village, first, from the perspective of the physical outcome of CCCL marketization, through the land consolidation project, the Duoli Company paid CNY 80 million for 330 mu of CCCL, which was used for the construction of new houses. The company resettled more than 600 households (more than 1800 villagers). The villagers who participated in the project received 40 m$^2$ of resettlement housing and a resettlement fee of CNY 35,000 per person. Moreover, the villagers contributed their land to set up a cooperative. In doing so, they could share in the land rent (2275 CNY/mu) and earn money by selling trees and flowers and working for the Duoli Company. By the end of 2016, approximately 312 local villagers were employed in service management, gardening, and construction, with a total salary of more than CNY 15 million. In 2017, through the project, the villagers' average income increased by CNY 27,000. Second, from the perspective of the distribution effect, the total revenue of the CCCL marketization was CNY 13,540,400. The government received CNY 2,031,100. Apart from the money given to the government by the village collectives, the remaining revenue was CNY 11,509,300. The villagers shared 20% of the revenue, which was CNY 2,301,900, and the village collectives shared 80% of the revenue, which was CNY 9,207,400. Third, from the perspective of the process efficiency, Baiyun Village used connections within the village, which improved the efficiency of the CCCL marketization process to a considerable extent. For example, when discussing the resettlement compensation and construction of new houses, though the opinions of the villagers differed, the village committee leaders and elites negotiated with all the stakeholders until they reached a consensus and a collective action was decided.

This coordination mechanism can solve the problem of information asymmetry, increase the effectiveness of the communication between different stakeholders, and reduce the conflict costs caused by the considerable heterogeneity of the stakeholders' preferences.

**Table 6.** Comparison of the CCCL projects between Baiyun Village and Huilong Village in terms of the performance differences.

| Village | Baiyun Village | Huilong Village |
|---|---|---|
| Marketization governance modes | Self-organized mode | Government-led mode |
| Physical outcome | Resettled 600 households (more than 1800 people) 40 m$^2$ of housing per person and compensation of 3500 CNY/person<br>Increase in the revenues of collectives and villagers<br>Creation of jobs for 312 villagers<br>Increase in personal income by 27,000 CNY/person | Improved infrastructure<br>Increase in the revenue of collectives by CNY 1,323,000<br>Increase in the revenue of villagers by CNY 984,000<br>Creation of jobs for more than 500 villagers |
| Distribution effect | Total revenue: CNY 13,540,400<br>Government: CNY 2,031,100<br><br>■　Collectives: CNY 9,207,400<br>■　Villagers: CNY 2,301,900 | Total revenue: CNY 3,390,000<br>Government: CNY 153,100<br>Huilong village:<br>■　Collectives: CNY 306,100<br>■　Villagers: CNY 306,100<br>Luohuatun Village:<br>■　Collectives: CNY 876,000<br>■　Villagers: CNY 584,000<br>Qunxing Village:<br>■　Collectives: CNY 140,900<br>■　Villagers: CNY 93,900 |
| Process efficiency | Project period: three years<br>No conflict<br>No holdout problem<br>No decision deficiency | Project period: one year<br>No conflict<br>No holdout problem<br>No decision deficiency |

For Huilong Village, first, from the perspective of the physical outcome of CCCL marketization, which was led by the local government, a marketization reform working group at the township and village levels was established. The entire marketization process was dominated by a top-down administration. The local government was responsible for designing the specific marketization transaction rules and negotiating the compensation with the stakeholders, land reclamation, and trade of the quota. Through marketization via the government-led mode, the revenue of the collectives in Huilong Village, Luohuatun Village, and Qunxing Village increased by CNY 1.323 million and that of the villagers in the three villages increased by CNY 984 million. Second, from the perspective of the distribution effect, the revenue from the marketization was CNY 3.39 million and the cost was CNY 2.624 million in total. The government received CNY 153,000, the collectives in Huilong Village received CNY 306,100, and each villager received CNY 92.3. The collectives in Luohuatun Village shared CNY 876,000, and the villagers shared CNY 584,000. The collectives in Qunxing Village shared CNY 140,900, and the villagers shared CNY 93,900. Third, from the perspective of process efficiency, the government-led model forced stakeholders to cooperate through an administrative power, which reduced the probability of opportunistic behaviors, such as holdouts. With the local government's strict control, the procedure of the trade of the quota was smooth and no conflicts emerged during the marketization process. However, the problem of information asymmetry and the decision-makers' bounded rationality remained unsolved. Although the government-led mode decreases ex ante and in-process costs to a large extent, it can generate a considerable amount of ex post costs, which can lead to low-efficiency decision-making and a decline in process efficiency.

## 5. Discussion

Owing to the rapid economic development and the large number of elites in the village, Baiyun Village had a chance to use a new governance mode that was completely different from the government-led mode that was adopted by most of the pilot areas. Based on the conceptual framework and comparison between the two governance modes, this study aimed to determine the influencing factors for, and performance differences in, the governance modes for CCCL marketization in rural China. In the preparation stage, complexity and uncertainty II are the key transaction attributes. The large number of stakeholders and their different preferences make negotiations difficult and relatively long. Stakeholders may engage in opportunistic behaviors, which can lead to failure before the other CCCL marketization stages can be reached. A land consolidation project needs specialized equipment and experts with professional knowledge. Such experts are responsible for the scheme design to ensure the success of the land consolidation project. In addition, one of two parties may terminate the contract, thereby causing the land consolidation to fail. We found that the two modes outsourced the land development, reclamation, and consolidation project to a specialized company. In the trading stage, specificity is the key attribute. The specific transaction procedure requires specialized staff, and those who are untrained are not qualified for the job. In other words, human specificity matters in the transaction stage. In the revenue distribution stage, frequency is the key attribute. In addition, the number of CCCL marketization stakeholders is large. Government staff must negotiate with each stakeholder, which can lead to a high frequency. In summary, the most appropriate governance mode for conducting CCCL marketization does not exist. Each governance mode has its own characteristics. For example, the self-organized mode performs better in dealing with uncertainty-related transactions, whereas the government-led mode performs better in handling frequency- and specificity-related transactions.

First, this study shares several similarities and differences with the studies of Wang and Tan and Tan and Heerink [1,35]. Wang and Tan deconstructed rural renewal into four stages: scheme design, investment and financing, implementation, and revenue distribution, which is similar to the CCCL decomposition in the present study. However, when the authors analyzed the transaction attributes of rural renewal, they did not consider the frequency, which is a key transaction attribute according to TCE. Moreover, unlike a general analysis of the effect of specificity on transaction costs, in this study, the transaction attribute of specificity was further divided into asset specificity and human capital specificity. In contrast to the present study, the aforementioned authors constructed an Institution of Sustainability framework and considered the influence of actors on the choice of a governance structure. The conclusions of this present study are consistent with those of the aforementioned research. Similar to the present study, Tan and Heerink deconstructed land readjustment into four stages: land acquisition and assembly, financing, demolition and construction, and revenue distribution. When the authors compared two cases, they considered the policy and legal environments. In other words, they analyzed the influence of the institutional environment on the choice of a governance structure. Furthermore, in terms of the governance structure, the two cases selected by the authors belonged to the hybrid structure. However, in the present study, the self-organized mode (hybrid) was compared with the government-led mode (hierarchy). Our conclusion resembles the aforementioned authors' conclusions, which is that differences in governance modes, natural resource endowments, socioeconomic development levels, and transaction costs, as well as the (mis)fit between the governance structure and transaction attributes, determined the performance of the governance modes for CCCL marketization.

Second, in this study, the transaction attributes, such as specificity, uncertainty, and frequency, are believed to have an influence on transaction costs. However, Shahab et al. pointed out that, apart from transaction attributes, the number of agents; transactor characteristics, such as past experiences, opportunism, trust between parties, and common preferences; policy characteristics, such as social connectedness, simplicity, age, and pre-

cision; and public participation have an effect on the transaction costs of the land use policy instrument [53]. Specifically, Shahab et al. classified the transaction costs related to the timing of the planning evaluation into three categories, namely ex ante transaction costs, ongoing transaction costs, and ex post transaction costs, from the perspective of the planning process [54]. In addition, Shahab et al.'s estimation of the magnitude and distribution of transaction costs in development rights transfer programs was based on interviewing landowners, developers, local authorities, and intermediaries, and classifying transaction costs into time-related costs and direct monetary expenses provides a reference for forthcoming studies, e.g., on the estimation of transaction costs for CCCL marketization [55].

Third, for the government-led mode, the local government has the final say on the decision of the company that is willing to use the land, as well as where, when, and how the CCCL will be transacted. Meanwhile, for the self-organized mode, the local community (collective economic organization) is comparatively powerful and has numerous connections, which can weaken the local government's interference to a certain degree. Therefore, it has the right to decide the CCCL area and use. CCCL marketization requires a substantial amount of money, which is not the case in most western and central villages in rural China. The lack of money makes CCCL marketization impossible in such villages. Therefore, most cases are promoted and guided by the local government, as it has the advantage of being able to provide the money needed for land consolidation, among other actions. In summary, owing to the aforementioned factors, the self-organized mode is not a more widespread governance structure than the public mode.

Fourth, the two cases were insufficient for completely determining the influencing factors for, and performance differences in, the governance modes, which is the major deficiency of a case study. However, it can provide a perspective on how the government-led and self-organized modes function during the marketization of CCCL, which is significant to the upcoming CCCL marketization in other regions in rural China. Additional cases should be examined to obtain a general conclusion, which we will improve upon in our forthcoming research.

Fifth, transaction costs of the two CCCL projects in this paper can only be compared in an indirect way. In reality, transaction costs are difficult to be defined and they cannot be calculated accurately in a direct way. If the transaction costs can be calculated, then the preferred governance mode for CCCL marketization can be selected based on the principle of transaction economizing. This is the reason that case comparative analysis, namely a qualitative assessment of transaction costs, is the methodology for most of the literature concerning TCE.

## 6. Conclusions

In this study, based on TCE, transaction stages and attributes and their alignment with governance modes were analyzed, and an analytical framework was established to assess the influencing factors for, and performance differences in, the two governance modes for CCCL marketization. The conclusions are presented below.

First, the transaction attributes in the three CCCL marketization transaction stages differed. Asset specificity, uncertainty I, uncertainty II, complexity, and frequency are the main transaction attributes in the preparation stage. Human capital specificity, uncertainty II, and frequency are the main transaction attributes in the trading stage, and frequency and uncertainty II are the main transaction attributes in the revenue distribution stage.

Second, the government-led and self-organized modes are aligned with the transaction attributes. The influencing factors for governance modes for CCCL marketization include complexity, asset specificity, uncertainty I, uncertainty II, and frequency. The property rights problem, the heterogeneity of stakeholders' preferences, and conservative ideas result in high complexity. When complexity is high, adopting a governance structure with decentralization characteristics, such as the self-organized mode, to coordinate diversified preferences and reduce the transaction costs caused by bargaining is suitable. When

asset specificity and uncertainty II are high, adopting a centralized governance structure, such as the government-led mode, to coordinate conflicts and reduce the probability of opportunistic behaviors and negotiation costs is suitable. When the frequency is high, adopting a centralized governance structure to reduce the transaction costs caused by bargaining between stakeholders is appropriate.

Third, the marketization performance of the two governance modes shares similarities and differences. From the perspective of the physical outcome, the government-led mode and self-organized mode can improve infrastructure and living standards and solve the unemployment problem of local villagers. From the perspective of the distribution effect, through the self-organized mode, Baiyun Village achieved a sustainable increase in revenue. Through the government-led mode, though the collectives and villagers in Huilong Village shared the revenue from the marketization, it was a one-time and unsustainable increase. From the perspective of process efficiency, the self-organized mode uses the leadership of village committees and elites. With administrative power, the government-led mode can force stakeholders to cooperate with one another. Furthermore, the two modes can solve conflicts, avoid the holdout problem, and improve the efficiency of the CCCL marketization process.

An important implication of this study is that diverse governance modes are necessary to promote CCCL marketization performance. The government-led mode in Huilong Village and the self-organized mode in Baiyun Village fit well with the transaction attributes. The former used the advantages of coercion to force relevant interest groups to cooperate with one another during the marketization. Meanwhile, the latter used connections within the village to reconcile stakeholders. The two modes decreased the transaction costs and enhanced the efficiency of land allocation. The evidence presented in this paper may be insufficient to illustrate the influencing factors for, and performance differences in, the governance modes for CCCL marketization. However, this study aimed to provide some ideas for improving related policies that can enhance CCCL resource governance in rural China. Substantial empirical evidence to prove the applicability of TCE in the public domain has yet to be found. Thus, additional case studies on various governance modes in different pilot areas should be conducted. Only in this way can excellent CCCL resource governing performance be realized.

**Author Contributions:** Z.C. is responsible for leading the research group to conduct the survey in two case areas. Y.Z. mainly focuses on designing the survey, interview, and questionnaire scheme, building the conceptual framework, and improving figures and tables. G.L. and Z.X. are key members of the research group and concentrate on summarizing the information during the survey, interview, and questionnaire. All authors have read and agreed to the published version of the manuscript.

**Funding:** This research has received financial support from the National Ten Thousand Talented Youth Project (2019) and the Chinese National Foundation of Social Sciences through project no. 16ZDA020. This research was also supported by the Zhejiang Provincial Natural Science Foundation of China under grant no. LQ21G030003 and by the Fundamental Research Funds for the Provincial Universities of Zhejiang (grant no. SJWY2021002).

**Data Availability Statement:** The data presented in this study are available on request from the corresponding author. The data are not publicly available due to privacy.

**Acknowledgments:** The authors of this paper would like to thank all the staff members in the Bureau of Natural Resources and Planning of Meitan County and Pi County for giving help during the survey. In addition, Rong Tan contributed a lot to guiding and improving this manuscript.

**Conflicts of Interest:** The authors declare no conflict of interest.

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
