# Peer review of "What Drives Different Governance Modes and Marketization Performance for Collective Commercial Construction Land in Rural China?"

_land, doi:10.3390/land10030319_

Round 1

Reviewer 1 Report

The revisions addressed the issues raised during the first round review. It is recommended to publish after the document clean up.

Author Response

Thank you for your comment.  I will make the document clean up before its publication.

Reviewer 2 Report

Having read this article twice, below are my feedback.

The introduction makes an interesting reading because of the brief description of China’s public property system in contrast to the private-oriented system in the Western Worlds. Generally, the discussion of China’s unique dualistic system provided a critical understanding of the basis for the  collective commercial construction land (CCCL) reform. I really consider the introduction to be well-detailed.

Theoretical analysis framework. I think the Figure 1 should be put within texts (maybe right before/after the reference to the Figure in the text. Also, the term “theoretical analysis framework” sounds misleading as it denotes both a theoretical and analytical framework and appears to be none of those two. Maybe the authors should consider simply renaming it a conceptual framework – I say this because it builds on concepts that directly link to CCCL and strives to answer research questions (and not building or embedding theories).

Watch your language? The authors say, “First, by decomposing CCCL, this study develops a 13 theoretical framework from the perspective of transaction cost economics.” Do you mean by “deconstructing…” I think you deconstructed and not decomposed.

Methodology: The article lacks a commanding methodological section. I think the authors can rebuild the section 2.4 into a methodology. Where they can clearly and fully tell readers about the process of the research and its methodological elements. The information is already there but needs firming up into a scientific narrative. Not having a well-structured methodological section is, to me, a key weakness of the paper.

The results and discussion speak well to what the objectives of the article is about.

Reviewer 3 Report

The subject of the paper is interesting, since it is important to know how the processes of urban land transformation and city development are managed in the different countries of the world.

However, this study shows serious weaknesses from a methodological and applicative point of view.

The methodological part about the marketization process, where the problem is discomposes into phases and factors is good, and the hypothesis of comparing the performances of self-organized and government-led modes on the basis of the elements identified in fig. 1 can provide useful considerations for scholars.

The second part of the method should be the one in which the two marketization modes are compared, and their performances are assessed. The authors declare in the abstract and in lines 352-364 that they conducted three type of interview and a close questionnaire.

However, apart from this quote, the explanation of the methodology is totally absent. The authors do not explain what the interview or questionnaire schemes are, what kind of information was obtained, how it was aggregated (especially considering that the questionnaires or interviews were different according to the subject), if the same item was differently assessed depending on the subject, etc.

Another methodological weakness is the lack of explanation of the choice of the two case studies.

It is clear that one case for each type of management was chosen. But it has not been explained why the Huilong village can be considered a representative case of all the other government-led mode cases. The same for Baiyun village.

Moreover, are they comparable to each other? If the answer is yes, from what points of view?

Minor comments

The whole paper should be revised because the sentences are often too short and the speech is too fragmented.

In the introduction (Lines 49-53 and 83-85) better explain the contents of the references.

Who made the assessments in Table 2? Are these the results of the interviews?

In conclusion, considering that there are serious methodological lacks, the paper has to be rejected in the current version. However, it is a good starting point for revising it and submitting a new version.

Round 2

Reviewer 2 Report

Having read this revised manuscript, I have a strong impression that it has been seriously improved in many ways – including renaming the Theoretical analysis framework" to "Conceptual framework" and improving on the messaging, and addition of section 3. Any other issues I observed are a matter of presentation and structure, and they are simply minor. Please note the following:

The section 2 refers to conceptual framework but do not presented as an independent section. It is possible that this is a problem caused by conversion from MS-word to PDF but it needs to be rectified. See lines 163 (p.4).

I would advise the authors against starting a section with a figure instead of text (see section 3, p.8). Readers should be guided by a text first before a graphic to enable them to gain a better understanding of the message, while considering the graphic as a reference material for in-depth understanding.

Author Response

Thank you for your comments. “Section 2 Conceptual Framework” is presented as an independent section to avoid the layout issue (L 164, p. 4). Figure 2 is put at the end of Section 3.1 to enable readers to gain a better understanding of the message (L 405, p. 10).

Reviewer 3 Report

The authors made all the changes or additions requested. I am fully satisfied with the improvements made to the presentation of the methodology, in particular paragraphs 3.2 and 3.3.

The manuscript can be accepted in the current form.

(just check line 163, heading 2 Conceptual Framework).

Author Response

Thanks for your comments. I have rectified “heading 2 Conceptual Framework” to avoid the layout issue (L 164, p. 4).

This manuscript is a resubmission of an earlier submission. The following is a list of the peer review reports and author responses from that submission.

Round 1

Reviewer 1 Report

This paper studies the two collective commercial land development modes in China. The research background review is solid, the methodology is well justified, and the data analysis and results are well presented. It is highly recommended for publication.

Three minor revisions need to be made before final publication:

  • it is recommended to add a short description of land right and land use right in the introduction to help readers understand the collective decision-making approach in China
  • keep number precision to 2 digits in the 3.1 section and the figure 2 legend
  • use a footnote or brackets to indicate "mu" as the land size unit